# An In Situ FTIR Study of DBD Plasma Parameters for Accelerated Germination of *Arabidopsis thaliana* Seeds

**DOI:** 10.3390/ijms222111540

**Published:** 2021-10-26

**Authors:** Alexandra Waskow, Lorenzo Ibba, Max Leftley, Alan Howling, Paolo F. Ambrico, Ivo Furno

**Affiliations:** 1Swiss Plasma Center (SPC), École Polytechnique Fédérale de Lausanne (EPFL), CH-1015 Lausanne, Switzerland; lorenzo.ibba@epfl.ch (L.I.); max.leftley18@imperial.ac.uk (M.L.); alan.howling@epfl.ch (A.H.); ivo.furno@epfl.ch (I.F.); 2CNR, Istituto per la Scienza e Tecnologia dei Plasmi, Sede di Bari, Via Amendola 122/D, 70126 Bari, Italy; paolofrancesco.ambrico@cnr.it

**Keywords:** plasma, *Arabidopsis thaliana*, FTIR, germination, reactive oxygen and nitrogen species, surface DBD

## Abstract

Current agricultural practices are not sustainable; however, the non-thermal plasma treatment of seeds may be an eco-friendly alternative to alter macroscopic plant growth parameters. Despite numerous successful results of plasma-seed treatments reported in the literature, the plasma-treatment parameters required to improve plant growth remain elusive due to the plethora of physical, chemical, and biological variables. In this study, we investigate the optimal conditions in our surface dielectric barrier discharge (SDBD) setup, using a parametric study, and attempt to understand relevant species in the plasma treatment using in situ Fourier transform infrared (FTIR) absorption spectroscopy. Our results suggest that treatment time and voltage are key parameters for accelerated germination; however, no clear conclusion on causative agents can be drawn.

## 1. Introduction

Current agricultural practices are not sustainable, as they deplete natural resources through deforestation, soil depletion, and intensive water use, and they have negative environmental consequences due to their heavy reliance on fertilizers and pesticides. Non-equilibrium plasma treatment for seeds may be an alternative to the latter because it produces no toxic residues, consumes little energy, and has low penetration depth, avoiding cell injury while supporting seed development.

Plasma treatments can modify the seed surface hydrophilicity or hydrophobicity to minimize water consumption or delay germination in non-ideal conditions. They can reduce the level of non-pathogenic and pathogenic microorganisms, as well as their toxins on the surface of the seed or plant, or, alternatively, degrade pesticides to less toxic compounds for remediation purposes. They can simultaneously influence plant growth parameters, such as germination and crop yield, as well as resistance to (a)biotic stresses, when dosed adequately [1].

Cold plasma is a weakly ionized gas, a complex mixture of electrons, ions, UV, and reactive species, which can exist at room temperature. The particular advantage of non-equilibrium plasmas is that the electrons are heated to very high temperatures (several thousand degrees centigrade) by acceleration in the electric fields of the discharge. In collisions, these high-energy electrons can transfer almost all of their energy to molecular internal energy, causing high-energy chemical reactions without strongly heating the gas, since the small kinetic energy transfer fraction is of the same order as that of the mass ratio. Thus, non-equilibrium plasma creates reactive species far beyond the energy range of conventional thermal chemistry, which is the principal interest of plasma chemistry.

Very often, dielectric barrier discharges (DBDs) at atmospheric pressures are used. The main distinguishing feature of a DBD is a dielectric layer made of glass, ceramic, plastic, or quartz. The dielectric is necessary to separate the electrodes, and to prevent arcing between exposed metallic electrodes which would result in thermal plasma and melting. A surface DBD (SDBD) has an electrode on each side of a dielectric, and the plasma is formed in gas at the edges of the high-voltage patterned electrode. The plasma duration and duty cycle can be chosen to maintain a sufficiently low-electrode temperature, which is advantageous when working with heat-sensitive biological substrates. 

In this study, we use a SDBD plasma to treat *Arabidopsis thaliana* Col-0 seeds, a plant model organism, to determine if an effect on germination can be observed. Despite the numerous successful results reported in the literature, it remains unclear which plasma treatment parameters are required to affect plant growth parameters because of the plethora of physical, chemical, and biological variables. Here, we find the optimal plasma conditions for germination in our setup using a parametric study, and to understand which species are relevant in the plasma treatment by using in situ Fourier transform infrared (FTIR) absorption spectroscopy.

## 2. Results

### 2.1. Germination Rate of Plasma-Treated Arabidopsis thaliana Col-0 Seeds

Germination rates were measured for parametric scans of plasma-treatment time, SDBD voltage, air flow rate, seed-to-SDBD distance, and AC excitation frequency. The default operating parameters were a plasma treatment time of 60 s; peak-to-peak voltage 8 kV; 2 L/min flow rate of dry synthetic air; seed substrate distance of 3.7 mm; and 10 kHz frequency with a power on/off modulation at 500 Hz and 10% duty cycle, corresponding to a burst of 2 cycles per modulation period. The 10% duty cycle was used to avoid heat shock of the seeds [2]. Figure 1 shows the germination rate of seeds measured at 48 h, compared to the control seeds with no plasma exposure.

As marked by the asterisks in Figure 1, we observed that scans of treatment time or voltage yielded statistically significant increases in the germination rate for 20, 60, and 80 s times, and for 6.5, 7.5, 8.5 kV_pp_ voltages. In contrast, modifying the flow rate, gap distance, or frequency made no significant changes, although no plasma treatments caused the germination rate to fall. The longest plasma-treatment time of 80 s had the highest statistical significance for the germination acceleration among all the combinations tested; suggesting the plasma duration was too short, perhaps due to the low 10% duty cycle, and further improvement could be achieved with longer plasma-treatment times.

Compiling the data across the five experimental scans in Figure 1, the mean germination rate of the five triplicate control groups (no plasma treatment) was 70.4% ± 8.1% and the mean germination rate for the five triplicate default parameter sets was 79.8% ± 9.0%. The standard deviation error bars are indicative of the natural variance of the seed germination results, and the statistical significance for the effect of the default plasma on the germination rate is represented by *p* = 0.0001. The significance is high due to the averaging across a large number of samples.

### 2.2. FTIR In Situ Analysis of Plasmas Using Parametric Scans

Since we observed that particular plasma conditions yielded a statistically significant effect on germination, plasma chemistry was compared between these conditions to identify the reason for this effect. Although a previous study used five surface analysis techniques on plasma-treated seeds, including SEM images [3], the focus of this study was to analyze plasma chemistry. Typical absorption spectra in the mid-infrared range of 2500–500 cm^−1^, with 4 cm^−1^ resolution and 20 s acquisition periods are shown in Figure 2. These measurements were taken independently of the germination experiments, without seeds, however, under the same conditions using exclusively dry synthetic air. Molecules such as ozone (O_3_), nitrous oxide (N_2_O), and nitrogen dioxide (NO_2_) were monitored in situ and the absorbance spectra for different treatment times, voltages, and air flow rates are shown, as they had the clearest trends. The presence of CO_2_ in the spectrum was due to variations in the air outside the reactor, see Section 4.8.

In Figure 2, the ozone increases strongly with time, less strongly with voltage, and decreases with air flow rate, as could be expected. The germination rate in Figure 1 increases correspondingly with the time series for ozone but does not follow the ozone trends for the voltage, nor the flow rate series. Furthermore, the germination rate is highest at 80 s treatment time, corresponding to the highest concentration of ozone, although similar ozone levels were produced in other parameter combinations (not shown) without having the same effect on germination. 

Figure 3 shows that the time evolutions of N_2_O during the series of treatment time and flow rate are qualitatively similar to ozone. Therefore, neither ozone nor N_2_O are correlated with the germination rate. In summary, from the FTIR observation of these gas species, the apparent lack of correlation means that it remains unclear which agent influences the germination rate.

It is instructive to note that if the experiment had been designed to measure only the time series, it would have been trivial—but incorrect—to infer that the increase in gas species in Figure 2a was responsible for the increased germination rate in the time series of Figure 1. This demonstrates the importance of experiments designed for scans over several independent parameters, and not just one single variable.

Moreover, if only the time series results had been selectively reported in this work, then this biased selection would deceptively imply that the increase in gas species was responsible for the increase in the germination rate. This shows the importance of objective reporting of all correlations, whether positive, null, or negative.

## 3. Discussion and Conclusions

Few studies have combined FTIR and plasma-seed treatments although, to the best of our knowledge, this has not been done using in situ FTIR. Plasma diagnostics using in situ FTIR have been performed in dry and humid air [4,5,6], and in specialized gas chemistries [7,8,9,10], although independently of seeds and germination rate studies.

Germination rate is a relatively quick and convenient biological measurement, but it may not always be the appropriate indicator; field data are perhaps more relevant. For example, Koga et al. [11] show that the harvest mass of *Arabidopsis* increases through DBD plasma treatment. Indeed, short-duration plasma has a long-term beneficial effect on harvest mass [11,12], without showing strong changes in germination rates. Even if no improvement in germination were observed, it could still be possible that other changes in the seed development may occur after plasma treatment. Therefore, studies should make conclusions carefully, and additional measurements such as harvest mass or RNA sequencing can be used to confirm results.

Using FTIR, it is possible to measure species such as NO_2_, N_2_O, NO, CO_2_, HNO_3_, HNO_2_, water, CO, N_2_O5, and O_3_ in an air plasma. Kyzek et al. [13] and Tomekova et al. [14] measured the plasma chemistry of a DCSBD using either ambient air or synthetic air. Interestingly, ambient air conditions resulted in a predominance in NO_x_ species, but resulted in primarily ozone followed by N_2_O_5_, N_2_O, and HNO_3_ when using synthetic air. Our results are comparable in terms of the presence of similar species, except that in our study, ozone was the dominant molecule, likely because synthetic air was used, but potentially also due to power. The steady increase of ozone with time in the measurements of Figure 2a is consistent with the observation of Shimizu et al. [15] that ozone increases monotonically with time for SDBD power densities below 0.1 Wcm^−2^. As shown in Section 4.4, the SDBD power was always below 0.08 Wcm^−2^, therefore the experiments were clearly in ozone mode, before secondary reactions led to significant concentrations of NO_x_ species. N_2_O_5_ is reported for longer discharges [4,5], depending on conditions, and was thus not detected here. Furthermore, ambient air is often in the range of 40–50% RH, and it has been shown by authors such as Koga et al. [11] and Sarinont et al. [12] that humidity is an important factor in improving harvest. However, our results suggest it is possible to have some effect in dry conditions, although the FTIR spectra have fewer radical types (none with H) because of this. Future work will explore the differences in FTIR spectra upon the addition of seeds and varying levels of humidity.

Based on Wang et al. [16], Kyzek et al. [13], and Tomekova et al. [14], the presence of nitrogen oxides appears sufficient to trigger changes in germination. However, Tomekova et al. [14] discovered the greatest DNA damage while using pure nitrogen. Our results indicate clear changes in ozone, N_2_O and NO_2_ concentrations, but other nitrogen species were not detectable.

Ozone and NO_x_ species are known to affect seed dormancy and germination [17,18,19,20]. However, in this study, there was no clear conclusion as to whether any particular molecule was responsible for the accelerated germination. Therefore, the effect is perhaps not due to the gas chemistry, but rather to ions, electrons, or other species such as NO, which play a potential role if able to reach the seed surface. Kyzek et al. [13] detected NO using FTIR absorbance spectroscopy in higher-power plasma where NO_x_ species were dominant; the NO absorbance was far smaller than for NO_2_ and N_2_O, which would be consistent with the absence of NO in the spectra of Figure 2. Laser-induced fluorescence (LIF) is another diagnostic that can detect short-lifetime species such as O, NO, OH, N, and HO_2_, but as these are short lived and very reactive, they disappear a short distance away from the plasma [4]. There are multiple long or short-lifetime species candidates which may affect germination, and it is not yet clear which of these are responsible for the plasma effect. For short-lifetime species, it is best to use LIF because single-pass FTIR is at the detection limit and cannot measure primary reactive species (radicals, ions, or metastables). Preliminary LIF studies in the present setup confirmed the presence of NO within the first millimeter from the SDBD, which suggests that reactive, short-lifetime species are responsible, but not ions, since an indirect plasma treatment was used. Two key points should be emphasized: multiple diagnostics should be used to cross-check results within the same study, and our FTIR methodology should be adjusted for NO to no longer be under the detection limit. It may be possible that NO is partially responsible for the effect on germination and therefore, future studies should explore this.

Despite this, the FTIR parametric study improved our understanding of how each variable influences plasma chemistry. Overall, similar species were detected across all spectra, but it was useful to confirm trends such as an increase in reactive species with time, and a decrease in concentration with an increased flow rate. Cimerman et al. [21] showed there is an increase in ozone with an increase in voltage, which in agreement with our results, and Yuan et al. [22]. However, Liu et al. [23] increased voltage and decreased their air flow rate to shift from ozone to NO_x_ mode. With a higher air flow rate, in both instances, ozone decreased but N_2_O increased in their study and not in ours. Perhaps this did not occur in our study because the operating ranges were small, (6.5–8.5 kV in comparison to 16–26 kV), but this remark is important to note for those who work in a specific regime. It may be possible that operating with such high voltages resulted in an elevated temperature and this contributed to a higher concentration of N_2_O. Chen et al. [24] showed that higher electrode temperatures suppress the generation of NO_2_ and promote the N_2_O generation. Likewise, it is known that increasing temperatures can decrease ozone concentration [25]. It is noteworthy that most energy is dissipated as heat in the system and only a fraction of energy is used to generate reactive species, as shown by Abdelaziz et al. [26] and therefore, power will partially determine the efficiency of reactive species generation. Our DBD operates in very low time-averaged power ranges of approximately 3–3.5 W, principally because of the 10% duty cycle, as shown by the Lissajous figure in Section 4.4, which may explain the dominance of ozone and the low concentration of N_2_O.

In conclusion, single-pass in situ FTIR measurement provides simultaneous monitoring of numerous biologically significant radicals during plasma-seed treatment and is potentially more relevant than monitoring downstream exhaust gas. However, it is too early to state if there are any different radicals or new behaviors observed using in situ FTIR. The experiments demonstrated the importance of performing scans over multiple independent variables to avoid erroneous interpretations. Currently, it remains unclear which agents influence germination. Further monitoring of reactive species would improve our understanding of how to tailor plasma-seed treatments.

## 4. Materials and Methods

### 4.1. Seed Material

*Arabidopsis thaliana* Col-0 seeds were cultivated in a plant chamber room and harvested in May 2019 using seeds from the Department of Plant Molecular Biology at the University of Lausanne. Therefore, the seeds at the time of the germination experiments were 18–20 months old. Seeds were stored in Eppendorf or Falcon tubes and kept at room temperature in the dark until used.

### 4.2. Germination Rate Measurements

Seeds were not sterilized nor subjected to seed preselection before plasma treatment. After plasma treatment, 30 seeds were sown immediately, or a few hours after (no later than within the same day), on water agar plates (20 g/L, using distilled water, pH of approximately 6.7) and kept in a phytotron (AR-36L2 PlantClimatics GmbH, Wertingen, Germany) under continuous light using Osram L 18W 77 G13 Fluora with a 24 h light cycle at 23 °C and 65% humidity. Germination was recorded at 48 h and seeds with roots were counted by eye. The germination rate was calculated as the number of seeds with roots divided by the total number of seeds and converted into a percentage.

### 4.3. Surface Dielectric Barrier Discharge Description

Figure 4a shows the airtight stainless steel reactor chamber, with an 18 cm diameter and 11 cm high, used to confine the SDBD air plasma and its gaseous products. The SDBD electrode assembly, shown in the photograph in Figure 4c from Sihon Electronics, comprises a high-voltage printed electrode in a stripe pattern on an alumina dielectric plate, placed on an aluminum ground electrode. The SDBD electrode power supply and high-voltage waveform are described in Section 4.4. The *Arabidopsis* seeds were placed on Teflon cylinders, several millimeters below the SDBD plasma, as shown schematically in Figure 4b. The infrared beam passes through the gap indicated by the dashed red line.

The feed gas system of a Bronkhorst mass flow controller provided the flow from a bottle of dry synthetic air (80:20 N_2_:O_2_) into the sidewall of the reactor as shown by the transparent tube in Figure 4a. The flow rates are given in Section 4.6. The exhaust gas flowed to the intake of a ventilation extraction system. 

### 4.4. SDBD Voltage Waveform and Lissajous Figure

The default AC voltage waveform was nominally a sine wave, 8 kV peak-to-peak at 10 kHz. To avoid overheating the substrate and seeds with continuous wave (cw) power, the AC supply was power modulated at 500 Hz with a duty cycle of 10%, similar to the technique used by Ambrico et al. [2]. This corresponds to a burst of two AC cycles per power modulation period as shown in Figure 5. The waveform is consistent and reproducible over time, although it is distorted by the high-voltage amplifier.

The SDBD voltage and the capacitor charge deduced from Figure 5 were used to measure the power dissipated in the plasma using the Lissajous figure method [27]. Since the 2 cycles are not identical due to the transient distortion by the high-voltage amplifier, the resulting Lissajous figure shown in Figure 6a is not conventional. The locus is not a parallelogram because the waveform is not a continuous sinusoidal wave, but a burst whose voltage converges to 0 every 2 cycles. This produces the discontinuity at the origin visible in the figure. Nonetheless, the area within the locus of the voltage vs. charge contour for each cycle represents the energy per cycle dissipated in the plasma. Accounting for the 500 Hz modulation frequency of the burst, the time-averaged dissipated power for different voltages is shown in Figure 6b. The power was calculated to be 1.2–3 W for nominal voltages ranging from 4 to 9 kV_pp_. If the entire striped area of the SDBD is taken into consideration, the corresponding power density is 0.03–0.08 W/cm^2^.

### 4.5. SDBD Temperature

The temperature was measured at the center of the SDBD with a FLIR E85 infrared camera and the measurements for 20 s, 60 s and 80 s were 28.1 °C, 31.2 °C and 31.8 °C, respectively, using 10% duty cycle, whereas using the original AC continuous power source provided with the SDBD was 38.8 °C, 52.8 °C, and 58.3 °C. The 10% duty cycle was selected for lower temperature plasma treatment to avoid overheating the seeds. The advantage of using pulsed power duty cycle control is that high voltages can be used to ensure plasma ignition, whilst nevertheless maintaining a low time-averaged power for ambient temperature experiments. High flow rates may also be useful to cool the seeds by convection, but this dilutes the gas species which are the active components of the plasma treatment.

### 4.6. Plasma Parameters for the Seed Treatment

The default parameters were 1 min delay with flow flushing before treatment; 10 kHz, 8 kV_pp_, 60 s plasma treatment time; 3.7 mm distance between seeds and plasma; and 2 L/min of dry synthetic air (80:20 N_2_:O_2_). The reactor’s total volume was 2.8 L with an internal gas volume of approximately 1.0 L. For the flow rates of 2, 4, and 5 L/min, the gas residence times were therefore 30 s, 15 s, and 12 s, respectively, which are of the same order as the treatment time of 20 s, 60 s and 80 s. Therefore, the time dependence of each species in Figure 3 is a complex convolution of its production rate by the plasma, secondary reactions in the gas or on surfaces, and its loss rate by convection in the air flow [4]. This result also depends on how long the seeds are left in the reactor after the plasma treatment.

The source voltage waveform for all excitation frequencies was a burst of 2 sinewave cycles with a 500 Hz on/off power modulation, provided by a Rigol DG4102 signal generator amplified by a Matsusada AMPS-20B20-LC (5m) power supply. At 10 kHz sinewave frequency, 2 cycles modulated at 500 Hz corresponds to a 10% duty cycle. Seeds were placed on Teflon spacers during treatment, or on ceramic plates to reduce the seed-plasma gap.

Humidity was measured using a Vaisala model HM42 probe and ranged between 1.5–3% RH. This low humidity is consistent with the use of dry synthetic air and only small outgassing of humidity from the reactor walls. Ozone measurements were taken independently and at a later time with an Eco Sensors model UV-100 ozone analyzer as the average of triplicates, and were used to confirm the trends observed with FTIR (data not shown). The highest ozone absorbance, in Figure 2a at 80 s, corresponds to 160 ppm at the time of measurement, however, there may be fluctuations in the values due to the aging of the DBD electrode.

### 4.7. Statistics

Differences between the two groups were assessed using ordinary one-way ANOVA. Each treatment group was compared to their respective control group, and bar graphs represent two independent experiments with 3 replicates each, for a total of 6 replicates. GraphPad Prism 9 (GraphPad Software, Inc., San Diego, CA, USA) was used for statistical analyses. All *p* values < 0.05 were considered to be significant and are shown in Figure 1. Only significant *p* values are shown in the graphs. A Welch two-sample *t*-test set to 95% confidence interval was used to compare control seeds with no plasma treatment to the default parameters of plasma treatment.

### 4.8. FTIR Single-Pass In Situ Diagnostic

The entire reactor, including gas flow control and electrical diagnostics, was mounted within the sample compartment of a Bruker Vertex 80 V vacuum Fourier transform infrared (FTIR) absorption spectrometer, as shown in Figure 7. The importance of in situ FTIR measurement is to avoid alteration of the gas during the recirculation of the exhaust to a downstream FTIR spectrometer. In situ measurement means that molecules are measured at their source, before any contact with surfaces, and instantly after their production without any transit delay when reactive species could be altered by secondary gas reactions. The FTIR spectra, therefore, represent the plasma species which would be directly in contact with the seeds. However, these preliminary in situ FTIR spectra do not appear to be any different from the conventional downstream FTIR measurements, showing that there are apparently no different species to be considered in the plasma treatment of the seeds with sufficiently high density to be detectable by this single-pass FTIR method.

The optics bench of the vacuum FTIR spectrometer is sealed by a 49.5 mm diameter KBr window on each side of the sample compartment. To protect these windows, and to confine the gaseous plasma products within the reactor, supplementary KBr windows of 22.5 mm diameter were installed on opposing reactor ports. The infrared beam, with a waist diameter of approximately 1.5 cm, passed through these windows (Figure 7), and between the SDBD and seed substrate which are only a few millimeters apart (Figure 4). The penalty for in situ measurements, in this case, is the small solid angle which reduces the transmitted infrared intensity. Because of the electrical connections and gas tubing to the reactor, the sample compartment remained open, hence the IR beam also traversed the ambient air outside of the reactor over a short distance, as marked by the red arrows in Figure 7. Variations in the laboratory atmosphere were responsible for the spurious CO_2_ signal in the IR spectra of Figure 2.

During the plasma, a strong absorption peak continuously grew at approximately 1352 cm^−1^, with a proportionately much smaller peak at 833 cm^−1^. These peaks remained permanently after the plasma was extinguished, and they are due to an irreversible surface transformation of the KBr reactor windows. We tentatively attribute this to a form of KNO_3_ [28,29], most likely due to an attack from NO_x_ species produced in the plasma.

## Figures and Tables

**Figure 1 ijms-22-11540-f001:**
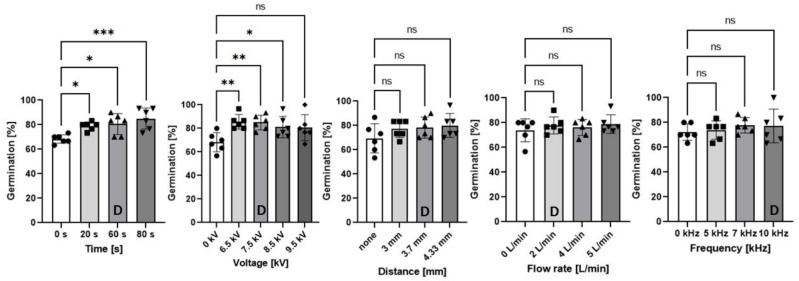
Germination rate results of plasma-treated *Arabidopsis* seeds from the parametric study. Germination was measured at 48 h with seeds grown on water agar in continuous light conditions at 23 °C with 65% RH. Experiments are an average of triplicates performed twice independently, for a total of 6 replicates. Asterisks denote statistical significance where * signifies *p* < 0.05; ** is *p* < 0.01; and *** is *p* < 0.001. The “ns” indicates “not significant”. The label “D” in each graph denotes the default parameter value.

**Figure 2 ijms-22-11540-f002:**
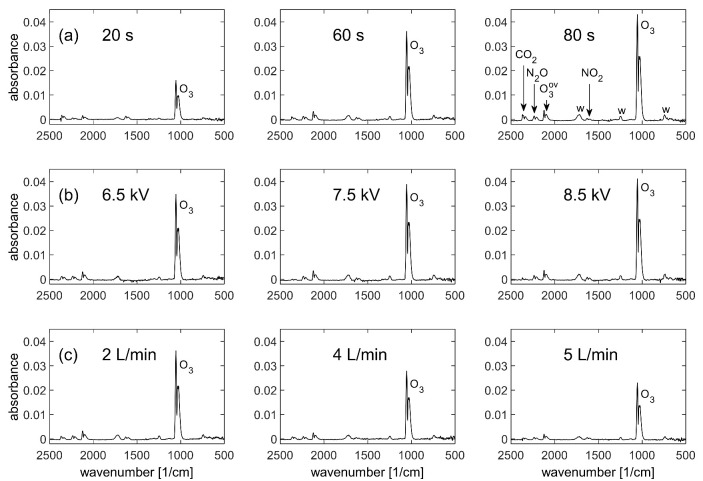
FTIR spectra of the plasma for different values of (**a**) plasma treatment time; (**b**) SDBD peak-to-peak voltage; and (**c**) air flow rate, plotted at the moment of maximum absorbance. Longer plasma-treatment time and higher voltage both increase the ozone concentration, whereas a higher flow rate decreases the ozone concentration, as expected. The species identified are labelled in the top right-hand graph; O_3_^ov^ indicates overtones of ozone, and “w” indicates non-gaseous compounds, which are tentatively attributed to surface reactions on the KBr windows.

**Figure 3 ijms-22-11540-f003:**
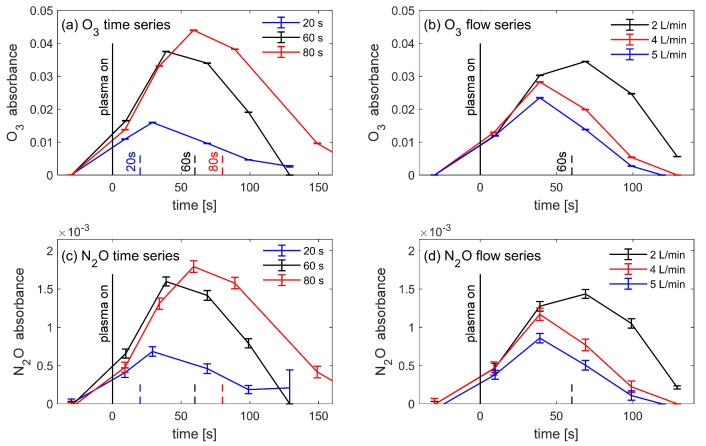
Time-dependent absorbance measurements of O_3_ and N_2_O from the series of treatment times (20, 60, and 80 s plasma duration) and flow rate (2, 4, and 5 L/min). The densities of both molecules increase with the plasma treatment time, and decrease with the air flow rate, as expected. The O_3_ and N_2_O molecules follow similar trends for each series. The measurements for the default parameters are shown in black.

**Figure 4 ijms-22-11540-f004:**
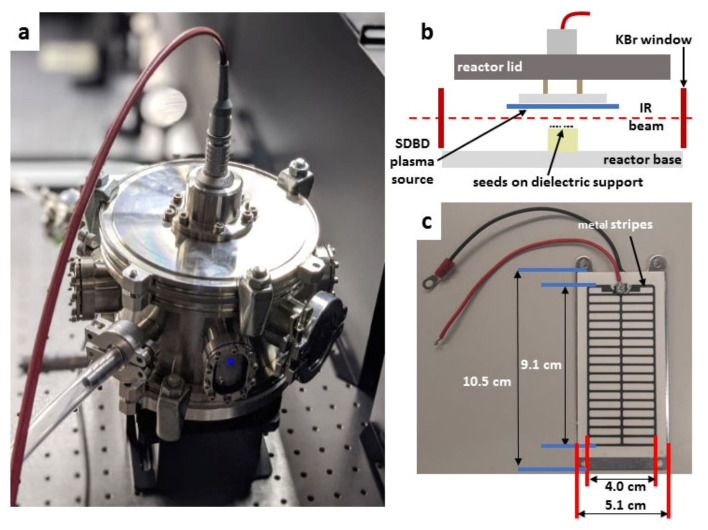
Surface dielectric barrier discharge (SDBD) plasma source enclosed in the plasma-seed treatment reactor. (**a**) Stainless steel reactor with high-voltage coaxial cable connection; (**b**) schematic of the interior with the inverted SDBD positioned above the seed substrate; (**c**) photograph of the high-voltage stripe SDBD electrode printed on an alumina dielectric plate. The ground electrode is an aluminum plate behind the dielectric.

**Figure 5 ijms-22-11540-f005:**
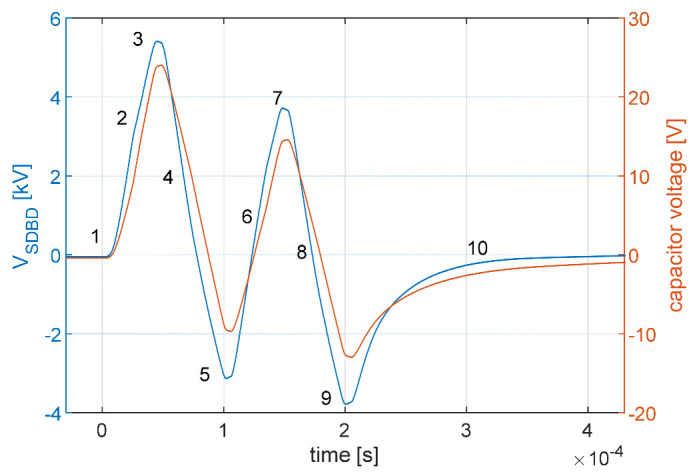
The voltage waveform of the SDBD plasma-seed treatment for nominal 8 kV peak-to-peak. The blue curve shows the voltage measured across the SDBD; the orange curve shows the voltage across the 68 nF series capacitor used to measure the charge on the electrode. The waveform distortion is caused by the transient response of the high-voltage amplifier when using high-frequency bursts. The numbers are used as references for Figure 6 to highlight specific key points.

**Figure 6 ijms-22-11540-f006:**
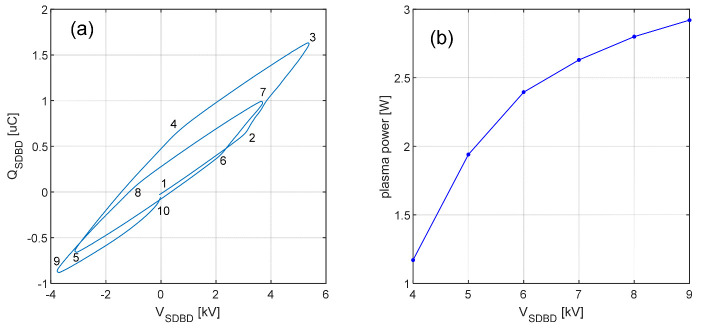
(**a**) Lissajous figure of the SDBD plasma for the two-cycle waveform at 8 kV_pp_ nominal voltage; (**b**) power calculation for nominal voltages 4–9 kV_pp_. The numbers are used as references for Figure 5 to highlight specific key points linked to the waveform.

**Figure 7 ijms-22-11540-f007:**
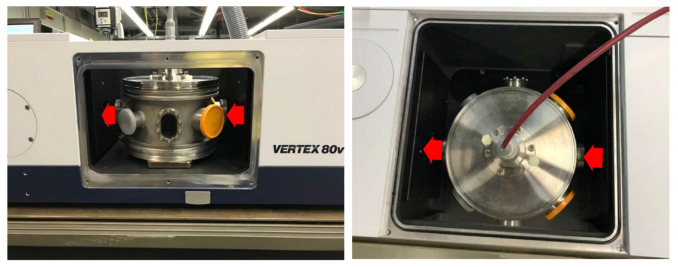
Side and top view photographs of the plasma reactor (see also Figure 4a) inside the sample compartment of the FTIR spectrometer, for in situ single-pass absorption measurements. The infrared passage through KBr windows is marked by the red arrows. The Faraday cage design of the reactor and its coaxial power supply cable protects against any possible electromagnetic perturbation to the FTIR instrument.

## Data Availability

The data presented in this study are available within the article and upon reasonable request.

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
