# Peer review of "An In Situ FTIR Study of DBD Plasma Parameters for Accelerated Germination of Arabidopsis thaliana Seeds"

_ijms, 2021, doi:10.3390/ijms222111540_

Round 1

Reviewer 1 Report

The authors present an in situ FTIR study of DBD plasma parameters for accelerated germination of Arabidopsis thaliana seeds. My comments are the following:

  1. It’s interesting and useful that the in-situ FTIR was used to follow the relevant species during seeds treatment by plasma. However, the in-situ FTIR using a plasma reactor (plasma catalysis, plasma polymer or plasma material interaction) is not a new story. More literature research are required related to in-situ FTIR plasma reactor (e.g 1-plasma depollution, https://doi.org/10.1002/ppap.201600114, https://doi.org/10.1038/srep31888, 2- plasma conversion, https://doi.org/10.1021/ie0606688, or 3-plasma polymer , Inc. J Polym Sci A: Polym Chem 36: 587–602, 1998)
  2. The information of plasma generated species in the gas phase is well studied under different conditions. However, it’s important to know the seeds' surface states which are the results of the reaction between the seeds and active species. To access the gas-solid interface (seeds- plasma generated species), the FTIR-Reflection Spectroscopy could be applied, instead of Transmission FTIR.
  3. Water is an important medium that could help the reaction between the seeds and plasma-generated species. I suggest making supplementary study in humid conditions to understand the role of the water ratio in the gas phase. Of course, the gas phase species could be monitored by your in-situ FTIR.

Author Response

Dear Reviewer,

Best regards,

Alexandra

Reviewer 2 Report

In this work, the authors study the effect of non-thermal plasma treatment on seed germination rate and try to correlate this with the reactive species generated in the gas-phase and detected by on-line FTIR measurements. They explore the dependence of many plasma parameters (treatment time, voltage, gas flow, plasma-to-target distance and frequency) on the seed germination rate and on the amount of reactive species produced. The conclusion of this work is that seed germination rate increase for higher treatment time and higher applied voltage and no clear correlation was found with the studied reactive species in the gas phase.

If on one side the approach of online FTIR measurements is very interesting and promising, the amount and quality of results presented is poor. The effect of plasma treatment on seed germination rate is very low and a dependence was found only for trivial parameters, like time and applied voltage. The reactive species that were studied are much likely not the ones that are responsible for the effect on the seed. I would expect that some of the primary reactive species (radicals, ions, metastables) have a major effect on the seeds, but FTIR is not the ideal method to detect/quantify these species, especially because their steady-state concentration is expected to be very low compared to the long/medium-lived species that were detected (ozone and nitrogen oxides).

To improve this work, that in my opinion currently is not suitable to be published on ijms, I suggest to find some experimental condition to emphasize the plasma effect on seed germination (longer treatment times, higher voltages, different seeds, …) and to identify the reactive species that are likely responsible for the effect. Using different feed gases could give some hints for example and trying with different techniques to quantify short-lived species.

Here some other minor comments to assess:

  • Some pictures of the seeds untreated and treated by plasma using the most effective conditions could be interesting for the readers.
  • It is not clear if the IR measurements with the seeds inside the chamber or not. If not, how do you explain the CO2 signal that you observe in the IR spectra?
  • Section 4.1 is not very clear, I suggest to rewrite it.
  • Some more details about the plasma source could be added in section 4.3, in particular regarding the feed gas system, the materials, the electrodes etc. Is the reactor air-tight?

Author Response

Dear Reviewer,

Best regards,

Alexandra

Round 2

Reviewer 1 Report

the manuscript could be accepted

Reviewer 2 Report

Dear authors,

thanks for your comments and for the modification you did on the paper.

If, on one side, I agree with you in saying that your results should be shared with the plasma agriculture community, on the other side, as I have already written in my first report, I don't think they are suffiently new/original or unexpected to justify publishing them in a medium/high IF journal as IJMS with a quite broad readership.

I suggest to consider some other journals more in the field of non-thermal plasma, where the results presented in this manuscript could be more appreciated.